# Impact of the *DRD2* Polymorphisms on the Effectiveness of the Training Program

**DOI:** 10.3390/ijerph19094942

**Published:** 2022-04-19

**Authors:** Katarzyna Świtała, Aleksandra Bojarczuk, Jacek Hajto, Marcin Piechota, Maciej Buryta, Agata Leońska-Duniec

**Affiliations:** 1Faculty of Physical Education, Gdansk University of Physical Education and Sport, 80-336 Gdansk, Poland; aleksbojar@gmail.com (A.B.); agata.leonska-duniec@awf.gda.pl (A.L.-D.); 2Laboratory of Pharmacogenomics, Department of Molecular Pharmacology, Maj Institute of Pharmacology Polish Academy of Sciences, 31-343 Krakow, Poland; hajto@if-pan.krakow.pl (J.H.); marpiech@if-pan.krakow.pl (M.P.); 3Institute of Physical Culture Sciences, University of Szczecin, 70-453 Szczecin, Poland; maciej.buryta@usz.edu.pl

**Keywords:** sports genetics, physical activity, *DRD2* polymorphisms, post-training effects, obesity-related traits, Caucasian women

## Abstract

Dopamine receptor D2 gene (*DRD2*) polymorphisms have been associated with cognitive abilities, obesity, addictions, and physical-activity-related behaviors, which may underlie differences in the effectiveness of training programs. What is not yet clear is the impact of *DRD2* polymorphisms on the effectiveness of exercise programs. Thus, the aim of this study was to investigate the association between the *DRD2* polymorphic sites (rs1076560, rs12364283, rs1799732, rs1800497, and rs1800498) and the body’s response to regular physical activity. We studied genotypes and haplotypes distribution in a group of 165 females measured for body mass and body composition measurements, lipid profile, and glucose levels before and after realization of a 12-week training program. When tested individually, statistical analyses revealed one significant genotype by training interaction under the general model (for the basal metabolic rate, BMR, *p* = 0.033). Carriers of the rs1076560 CC genotype exhibited a decrease in BMR in response to training (*p* = 0.006). Haplotype analyses also showed that (i) the CACCC and CACTT haplotypes were associated with a post-training decrease in glucose level (β = −4.11, *p* = 0.032; β = −6.86, *p* = 0.020, respectively); (ii) the CGCCT with an increase in BMR (β = 0.65, *p* = 0.003) and fat free mass (FFM, β = 1.20, *p* = 0.009); (iii) the CA-CT with a decrease in low-density lipoprotein cholesterol (LDL, β = −17.26, *p* = 0.046). These results provide some evidence that the *DRD2* polymorphisms may play a role in post-training changes in lipid and carbohydrate metabolism, and, as a consequence, in the effectiveness of training programs.

## 1. Introduction

Since the discovery of the physiological functions of dopamine (DA; 3,4-dihydroxy-phenylethylamine) in 1957 [1], this catecholamine neurotransmitter and its receptors have attracted the attention of many scientists from around the world. DA is synthesized from the amino acid tyrosine (Tyr), and generally acts on neuronal circuitry via a rather slow modulation of the fast neurotransmission that is mediated by glutamate and gamma-aminobutyric acid (GABA) [2,3].

DA in the brain is involved in numerous key central nervous system (CNS) functions, including motivation, feeding, stress tolerance, reward system, sleep regulation, attention, self-control, working memory, and learning [2,3]. In the periphery, DA plays essential physiological roles in the control of olfaction, retinal processes, cardiovascular functions, hormonal regulation, sympathetic regulation, and the immune system, among others [3]. Additionally, its impact on physical-activity-related behaviors was described in animal and human studies. More specifically, DA is engaged in the development of fatigue, which leads to a reduction in exercise intensity or its interruption, through the modulation of circuits linked to motor control and tolerance to heat stress, as well as the previously mentioned motivation and reward system [2,4,5,6].

DA acts through five distinct membrane receptors, which belong to the family of seven-transmembrane domain G protein-coupled receptors. Based on their structural, biochemical, and pharmacological properties, the DA receptors (DRDs) were divided into two subtypes: D1-like (DRD1 and DRD5) and D2-like (DRD2, DRD3, and DRD4). These receptors, respectively, stimulate and inhibit adenylyl cyclase, thereby regulating intracellular levels of cyclic adenosine monophosphate (cAMP) [7]. DRD2s are highly expressed in the striatum and pituitary gland. The deficiency of these receptors has been associated with decreased locomotor activity, increased prevalence of obesity, and the modification of the electrophysiological characteristics of DRD2-expressing neurons, among others [8,9,10]. Thus, these receptors play a key role at the postsynaptic level, in addition to acting as autoreceptors, in regulating of synthesis and release of DA [9].

The human *DRD2* gene is localized on the long arm of chromosome 11 at the q23.2 *locus*, and involves an area of 65.56 kb [11,12]. By the mechanism of alternative splicing, two main molecularly and functionally distinct isoforms (a short variant—D2S and a long variant—D2L) are generated [9]. Numerous single nucleotide polymorphisms (SNPs) are localized within the gene, which is important for the modulation of central nervous dopaminergic signaling. We selected five functional *DRD2* polymorphic sites (rs1076560, rs12364283, rs1799732, rs1800497, and rs1800498) which are considered to alter expression, splicing, and/or neuronal activity. Detailed characteristics of selected SNPs are presented in Table 1 [13,14,15,16,17].

In light of the above-mentioned findings suggesting that the *DRD2* variants are associated with cognitive abilities, obesity phenotypes, and physical-activity-related behaviors, among others, *DRD2* is a candidate gene related to the body’s training response. These associations are mainly confirmed in females [18]. However, the potential impact of the *DRD2* polymorphisms on the effectiveness of fitness programs is still unclear. Therefore, the study aimed to determine whether the selected *DRD2* polymorphic sites (rs1076560, rs12364283, rs1799732, rs1800497, and rs1800498), individually or in haplotype combination, would influence the post-training changes of selected body mass and body composition measurements, as well as biochemical parameters (lipid profile and glucose levels). To investigate the potential association between SNPs and physical outcomes, we assessed the genotypes and haplotypes distribution in Caucasian females taking part in a 12-week aerobic training program. Selected body mass and body composition, as well as biochemical parameters, were measured before and after the completion of the training program.

## 2. Materials and Methods

### 2.1. Ethics Statement

The experimental protocols were positively verified by The Ethics Committee of the Regional Medical Chamber in Szczecin (no. 09/KB/IV/2011 and 01/KB/VI/2017), and were conducted according to the World Medical Association Declaration of Helsinki and Strengthening the Reporting of Genetic Association studies statement (STREGA). Participants qualified for the research received an information sheet about the aim of the study, procedures used, benefits and risks, and gave a written consent form. Pseudonymization was used as the data protection method.

### 2.2. Participants

Caucasian women of Polish nationality (*n* = 165; age: 21 ± 1 years; body mass: 61 ± 2 kg; body height: 168 ± 2 cm) were selected for the study. The inclusion criteria were as follows:-low level of physical activity self-reported with the use of the Global Physical Activity Questionnaire;-no metabolic, neuromuscular, or musculoskeletal disorders;-refrained from using medications and supplements;-nonsmokers.

### 2.3. Dietary Program

The women participated in a dietary program, and were asked to keep a balanced diet based on their dietary plan, which was established during a nutritional appointment. The meeting included a recommendation and a prescription for a proper diet matched with nutritional status and individual energy needs. The average daily macronutrient ratio was recommended (expressed as a percentage of total calories): 45–65% from carbohydrates, 10–20% from protein, and 20–35% from fat (decreasing the intake of saturated fats, and increasing the intake of unsaturated fats). The participants were also advised to keep a daily cholesterol intake of less than 300 mg, with a minimum dietary fiber intake of 25 g. The women wrote down their daily food and drink consumption during the program. Their diet was assessed at weekly consultations.

### 2.4. Training Phase

The experimental training sessions were preceded by a week-long familiarization stage (3 training units, 30 min each, at ~50% of HRmax). Each proper training session included a warm-up (10 min), aerobic exercise (a combination of two styles, including high and low impact; 43 min), and a cool-down phase (breathing–relaxing exercise with stretching; 7 min). The high-impact style contained running, jumping, and hopping. The low-impact style included movements with at least 1 foot on the floor at all times. A 12-week program of low–high impact aerobics was divided as follows:-3 weeks (9 training units), 60 min each, at 50–60% of HRmax, tempo 135–140 BPM;-3 weeks (9 training units), 60 min each, at 60–70% of HRmax, tempo 140–152 BPM;-3 weeks (9 training units), 60 min each, at 65−75% of HRmax, tempo 145–158 BPM;-3 weeks (9 training units), 60 min each, at 65−80% of HRmax, tempo 145–160 BPM.

More detailed information on the training phase is presented by Leońska-Duniec et al. [19]. The adherence rate to the exercise program was 80%.

### 2.5. Body Composition Measurements

The selected body mass and body composition variables were measured before and after the realization of a 12-week training program. They were assessed using the bioimpedance method, which was performed using electronic scale Tanita TBF 300 M (Arlington Heights, IL, USA) as described by Leońska-Duniec et al. [19]. The following parameters were noted:-total body mass (BM; kg);-body mass index (BMI; kg/m^2^);-basal metabolic rate (BMR; kcal);-fat mass (FM; kg);-fat free mass (FFM; kg);-fat mass percentage (%FM; %);-total body water (TBW; kg).

### 2.6. Biochemical and Hematological Analyses

Fasting blood samples were obtained from the elbow vein in the morning, before and after the training program. The biochemical and hematological analyses were performed as described earlier [19], immediately after blood collection. The parameters obtained using the Random Access Automatic Biochemical Analyzer for Clinical Chemistry and Turbidimetry A15 (BioSystems S.A., Barcelona, Spain) were:-total cholesterol (TC, mg/dL);-triglycerides (TGL, mg/dL);-high-density lipoprotein cholesterol (HDL, mg/dL);-low-density lipoprotein cholesterol (LDL, mg/dL);-glucose (mg/dL).

### 2.7. Genetic Analyses

A GenElute Mammalian Genomic DNA Miniprep Kit (Sigma, Steinheim, Germany) was used for the extraction of genomic DNA from the buccal cells according to the manufacturer’s protocol. An allelic discrimination assay on a C1000 Touch Thermal Cycler (Bio-Rad, Feldkirchen, Germany) instrument with TaqMan^®^ probes was used to genotype all samples. To discriminate *DRD2* rs1076560, rs12364283, rs1799732, rs1800497, and rs1800498 alleles, TaqMan^®^ Pre-Designed SNP Genotyping Assays were used (Applied Biosystems, Waltham, MA, USA) (assay ID: C_2278888_10, C_31503501_10, C_33641686_10, C_7486676_10, and C_2601166_10, respectively). The assays contained primers and fluorescently-labeled (FAM and VIC) minor groove binder (MGB) probes to detect alleles.

### 2.8. Statistical Analyses

All statistical analyses were performed in R (https://cran-r.project.org, accessed on 18 October 2021, version 4.1.0). An HWChisq function from Hardy–Weinberg v. 1.7.4 R package was used to test for the Hardy–Weinberg equilibrium. No variants violating HW equilibrium were found. To check the influence of the *DRD2* rs1076560, rs12364283, rs1799732, rs1800497, and rs1800498 polymorphisms on training response, the mixed 2 × 2 ANOVA with one between-subject factor (genotype) and one within-subject factor (time: before training vs. after training) was used. Additionally, for a parameter with statistically significant interaction, a normality Kolmogorov–Smirnov test and one-way ANOVA for a simple main effect of each variable with Bonferroni correction as post hoc analysis were performed. Haplotype analysis was conducted with haplo.stats v. 1.8.7 R package and haplo.glm regression function. Percentage change overtraining was used as the dependent variable, whereas the *DRD2* haplotypes were used as the independent variables. The level of statistical significance was set at *p* < 0.05.

## 3. Results

All variants conformed to Hardy–Weinberg equilibrium (*p* = 0.946, *p* = 0.206, *p* = 0.183, *p* = 0.504, *p* = 0.674, for the rs1076560, rs12364283, rs1799732, rs1800497, and rs1800498, respectively). Table 2, Table 3, Table 4, Table 5 and Table 6 present the results of the analysis of the training responses by the *DRD2* genotypes using a mixed 2 × 2 ANOVA. The majority of studied parameters altered significantly during training; however, these changes did not differ concerning the *DRD2* genotypes. We found only one statistically significant genotype by training interaction under the general model (for the BMR, *p* = 0.033, Table 2). Carriers of the *DRD2* rs1076560 CC genotype exhibited a significant ~0.5% decrease in BMR in response to applied training (*p* = 0.006 with Bonferroni correction). However, under the dominant model, no significant genotype × training interactions were found.

Reconstruction of haplotypes revealed 8 haplotypes with frequency > 1%. The most common (a baseline haplotype) was CACCT (rs1076560, rs12364283, rs1799732, rs1800497, and rs1800498, respectively). We found a significant association of four haplotypes CACCC, CACTT, CGCCT, and CA-CT (Table 7). The CACCC and CACTT haplotypes were associated with a greater decrease in glucose (β = −4.11, *p* = 0.032; β = −6.86, *p* = 0.020, respectively) compared with the baseline haplotype. The CGCCT haplotype was associated with a greater increase compared with a baseline haplotype in BMR (β = 0.65, *p* = 0.003) and FFM (β = 1.20, *p* = 0.009), whereas the CA-CT was associated with a greater decrease in LDL (β = −17.26, *p* = 0.046).

## 4. Discussion

Systematic physical activity is one of the key factors for the prevention of lifestyle diseases, such as obesity. The number of people with overweight and obesity is growing worldwide; consequently, the prevention of weight gain is a very important health problem [19,20]. Currently, the difficulty is still in defining the detailed molecular mechanism of post-training changes in the human organism. Numerous components affect the body’s response to training, including individual predispositions, different types of physical effort and training, degree of training, age, gender, diseases, and others [17,21,22]. To answer the question of whether the *DRD2* polymorphisms influence effectiveness of the training program, we assessed the genotypes and haplotypes distribution described in five *DRD2* polymorphic sites (rs1076560, rs12364283, rs1799732, rs1800497, and rs1800498) in Caucasian females taking part in the 12-week training program. The changes of selected body mass and body composition measurements, as well as biochemical parameters (lipid profile and glucose levels) measured before and after the training, have been analyzed in the context of carrying the *DRD2* genotypes their haplotypes combinations.

Although the majority of studied parameters altered significantly during the 12-week aerobic training program, these changes did not differ concerning the *DRD2* genotypes. Only one statistically significant genotype by training interaction under the general model was found. Carriers of the rs1076560 CC genotype exhibited a decrease in BMR in response to applied training in comparison with individuals carrying AA and CA. Although the declines in BMR were significant, they were small (~0.5%), and may not have been physiologically relevant, especially because under the dominant model (where rare AA homozygotes were added to CA heterozygotes), no significant genotype by training interaction was found.

The animal studies demonstrated an association between DRD2 and both movement patterns and overall locomotor activity level. The DRD2 knockout mice showed decreased initiation of spontaneous activity in comparison to wild-type mice [8]. The higher expressions of DRD2 and DRD4 (19% and 24%, respectively) in mice selectively bred for high levels of physical activity than the controls were also shown [23]. In human studies, the associations between these genes and physical activity are mainly confirmed in women. In a study including a cohort consisting of 721 participants, Simonen et al. (2003) revealed that variation in the DRD2 gene was significantly associated with the level of sport participation and occupational physical activity among white women [18]. Another study performed on 900 Polish adults showed a lack of relationship between selected DRD2 polymorphic variants and the level of physical activity in men [24]. In addition, Lee et al. (2020) revealed that the effect of the DRD2 polymorphisms, particularly rs1800497, on females participating in sport is much greater in the younger population, suggesting that genetic influences on physical-activity-related behaviors reduce with age. They implied that adult physically active women are weakly affected by DRD2 because adult females are more exposed to behavioral and environmental factors that may influence physical-activity-related behaviors for a much longer period in comparison to adolescents [25]. This observation was confirmed by other studies, which have shown that adult females are more susceptible to the social environment than males [26]. Unfortunately, our study only included adult women (men did not report to the experiment); thus, we did not have the opportunity to compare the results between genders and different age groups.

The influence of the *DRD2* polymorphisms on the post-training effects is largely unknown. Organization of the experiment, consisting of careful regulation of both food intake and physical activity of a homogeneous population, is very difficult. Thus, only a few authors have tried to explain this problematic issue. In a study including 202 obese adults participating in a 1-year weight-loss program, Winkler et al. (2012) analyzed whether rs1800497 polymorphic site within *DRD2* was associated with body mass changes in response to applied training and diet. They revealed that younger hetero- or homozygous for the T allele (often referred to as A1+) participants showed higher BMI at baseline, and had problems in losing weight and maintaining weight loss during the experiment [27]. Additionally, Cameron et al. (2013) examined if the rs1800497 genotype was related to changes in body weight, energy expenditure, and food preference in 127 obese postmenopausal women taking part in a 6-month intervention including caloric restriction with or without resistance training. The carriers of the T allele lost significantly less body mass, BMI, and FM than the C allele carriers, and had increased carbohydrate intake in the group in which diet was connected with training [28]. However, our results did not confirm the relationship between rs1800497 polymorphism and post-training changes in body weight and composition. The reasons for the inconsistency of results may be due to factors such as the age of the participants, ethnical origin, too low initial BMI, or too short time of the experiment. We established only one association between the rs1076560 CC genotype and post-training decrease in BMR; however, the declines in BMR were small and may not have been physiologically relevant. The obtained results cannot be discussed because we did not find studies that assessed the influence of this polymorphism on the post-training effects.

When the obtained results were included in the complex haplotype analysis, the novel finding was that carriers of the CACCC and CACTT haplotypes displayed greater post-training effects, in terms of glucose level decrease, in comparison with individuals carrying the most common CACCT haplotype. This result suggests that harboring these specific haplotypes might be favorable for achieving the desired training-induced glucose level changes. Additionally, the CGCCT haplotype was significantly associated with an increase in BMR and FFM in response to applied training, when compared with a baseline haplotype. These results imply that some individuals might benefit from carrying the CGCCT haplotype, as regards the post-training FFM and BMR changes. Another observation was that carriers of the CA-CT haplotype displayed a greater post-training decrease in LDL, suggesting that harboring this haplotype might be beneficial for achieving the training-induced LDL level changes. Our results confirmed that methods based on haplotypes referring to multiple SNPs which are located closely together on the same inherited chromosome are more informative than methods based on individual SNP. The haplotypes analysis may provide additional power for mapping genes, and insight on factors affecting the dependency among genetic markers. Such results may give information important for understanding complicated interactions between numerous gene variants. Biologically, polymorphisms on a haplotype may cause several changes in coding, expression, or splicing, and, therefore, lead to a greater joint effect on the studied trait than the single change caused by SNP [29].

To the best of our knowledge, this is the first study to analyze the association of the *DRD2* rs1076560, rs12364283, rs1799732, rs1800497, and rs1800498 polymorphisms in haplotype combination with post-training changes of selected body mass and body composition measurements, as well as biochemical parameters in physically active participants. Therefore, the obtained results cannot be discussed with direct comparisons to other studies. However, Zhang et al. (2007) performed a detection of allelic mRNA expression in the human striatum and prefrontal cortex, and then performed SNP scans of the *DRD2 locus* and genotyped 23 SNPs. The analysis showed complicated interactions between *DRD2* variants that modify mRNA expression and splicing [13]. So far, many authors have analyzed the relationship between *DRD2* haplotype and alcohol, nicotine, or drug addiction; some types of cancer; and psychiatric disorders such as ADHD or schizophrenia; confirming that this type of analysis is more informative and provides new associations [30,31,32,33,34,35].

## 5. Conclusions

The obtained results provide some evidence that *DRD2* may play an important role in post-training changes of lipid and carbohydrate metabolism, and, as a consequence, in the effectiveness of training programs. The individual and complex haplotype analysis provided new information about the associations between training-induced glucose levels and lipid profile changes and the *DRD2* polymorphisms. More specifically, novel findings of the study were: (i) the rs1076560 CC genotype exhibited a small decrease in BMR in response to applied training, but only under the general model; (ii) the CACCC and CACTT haplotypes might be favorable for achieving the desired training-induced glucose level changes; (iii) the CGCCT haplotype might be favorable for achieving the desired training-induced BMR and FFM changes; and (iv) the CA-CT haplotype might be beneficial for achieving the training-induced LDL level changes. This preliminary data suggest that *DRD2* may be a promising molecular marker to predict the benefits from training programs or a physically active lifestyle; however, more studies are needed to establish precisely the *DRD2* gene × physical activity interactions. An understanding of the genetic background of training-induced body changes will allow us to clarify the conditions of physical activities for individuals. In the future, this knowledge may help to identify people who are expected to react well or poorly to exercise, which may help reduce the number of obese people, and improve their health.

## Figures and Tables

**Table 1 ijerph-19-04942-t001:** Characteristics of the studied *DRD2* polymorphic sites.

*DRD2* Polymorphisms			
Variant	Position in the Gene	Reported Functional Consequences at the Molecular Level	Reported Clinical Associations
rs1076560; A/C	Intron 6	Affects alternative splicing: the A allele reduces the formation of D2S in favor of D2L.	The A allele: - higher risk of alcoholism, drug abuse, non-small cell lung cancer, schizophrenia; - reduced performance in working memory and maintaining attention.
rs12364283; A/G	Promoter	Affects allelic mRNA expression: the G allele confers higher transcriptional activity.	The G allele: - increased risk of binge eating disorder, symptoms of schizophrenia, response to stressful situations, better working memory; - decreased risk of autism, obesity and insulin resistance, alcoholism, and drug addiction.
rs1799732; −141C Ins/Del	Promoter	Affects allelic mRNA expression: the C-del allele reduced promoter activity which results in decreased protein expression.	The C-del allele: - higher risk of overweight/obese; - antipsychotic-induced weight gain in schizophrenia.
rs1800497; TaqIA; C/T; Glu713Lys	10.5 kb downstream of *DRD2* in ankyrin repeat and kinase domain containing-1 gene (*ANKK1*)	Altered substrate bindingspecificity and D2R expression: AA and GA reduced D2R densities.	The AA and GA genotypes: - improvements in the working memory training program; - higher risk of obesity, alcohol, nicotine and drug addiction, emotional eating habits, and certain neuropsychiatric disorders.
rs1800498; TaqID; C/T	Intron 2	Affects allelic mRNA expression.	The T allele: - higher risk of autism spectrum disorders, schizophrenia, alcohol and nicotine addiction.

**Table 2 ijerph-19-04942-t002:** Training responses by *DRD2* rs1076560 genotypes.

Parameter	DRD2 rs1076560 Genotypes						*p* Values			
	AA (*n* = 6)		CA (*n* = 49)		CC (*n* = 110)		Genotype	Training	Genotype × Training	Genotype × Training AA + CA vs. CC
	Before Training	After Training	Before Training	After Training	Before Training	After Training				
Body mass (kg)	60.43 ± 6.08	58.8 ± 5.63	60.86 ± 7.59	60.17 ± 7.6	60.52 ± 7.89	59.8 ± 7.73	0.945	**0.000**	0.379	0.793
BMI (kg/m^2^)	21.7 ± 2.68	21.13 ± 2.48	21.37 ± 2.11	21.19 ± 2.06	21.71 ± 2.58	21.48 ± 2.54	0.747	**0.000**	0.228	0.983
BMR (kJ)	6081 ± 261.78	5926.67 ± 206.31	6082.18 ± 336.36	6053.41 ± 326.79	6043.97 ± 330.64	6009.3 ± 319.92	0.730	**0.000**	**0.033**	0.678
%FM (%)	24.5 ± 7.65	21.58 ± 7.02	23.62 ± 5.47	22.41 ± 5.97	23.93 ± 5.41	22.65 ± 5.48	0.956	**0.000**	0.207	0.752
FM (kg)	14.9 ± 5.78	13 ± 5.24	14.73 ± 5	13.84 ± 5.2	14.85 ± 5.16	13.94 ± 5.16	0.974	**0.000**	0.344	0.747
FFM (kg)	45.17 ± 2.04	46.02 ± 1.65	46.15 ± 3.32	46.43 ± 3.44	45.54 ± 3.23	46.01 ± 3.28	0.620	**0.005**	0.508	0.570
TBW (kg)	32.88 ± 1.64	33.67 ± 1.34	34.01 ± 2.77	34.06 ± 2.5	33.26 ± 2.55	33.7 ± 2.45	0.388	**0.042**	0.204	0.187
TC (mg/dL)	160.83 ± 21.05	155.33 ± 22.63	167.59 ± 28.79	166.51 ± 32.44	171.15 ± 23.01	169.38 ± 24.99	0.392	0.374	0.888	0.954
TGL (mg/dL)	81.5 ± 24.17	73.33 ± 25.66	71.39 ± 23.68	81.69 ± 35.49	84.16 ± 35.49	84.95 ± 35.72	0.268	0.842	0.173	0.173
HDL (mg/dL)	68.73 ± 26.52	62.52 ± 17.44	65.56 ± 13.22	59.67 ± 13.96	64.66 ± 12.44	61.52 ± 13.26	0.851	**0.002**	0.296	0.118
LDL (mg/dL)	75.67 ± 14.57	78.15 ± 11.16	87.69 ± 23.32	90.5 ± 27.92	89.57 ± 21.27	90.87 ± 22.01	0.298	0.457	0.904	0.655
Glucose (mg/dL)	77.67 ± 6.98	68.83 ± 8.33	79.16 ± 8.63	75.37 ± 9.19	77.88 ± 10.54	76 ± 10.58	0.561	**0.001**	0.178	0.143

Mean ± standard deviation; *p* values (ANOVA) for main effects (genotype and training) and genotype × training interaction; bold *p* values—statistically significant differences (*p* < 0.05).

**Table 3 ijerph-19-04942-t003:** Training responses by *DRD2* rs12364283 genotypes.

Parameter	DRD2 rs12364283 Genotypes						*p* Values			
	AA (*n* = 147)		GA (*n* = 16)		GG (*n* = 2)		Genotype	Training	Genotype × Training	Genotype × Training GG + GA vs. AA
	Before Training	After Training	Before Training	After Training	Before Training	After Training				
Body mass (kg)	60.62 ± 7.9	59.85 ± 7.79	60.61 ± 6.36	60.08 ± 6.17	60.45 ± 4.74	60.1 ± 4.53	0.998	0.174	0.801	0.516
BMI (kg/m^2^)	21.61 ± 2.51	21.39 ± 2.47	21.82 ± 1.86	21.53 ± 1.8	20.05 ± 0.78	19.95 ± 0.64	0.652	0.117	0.816	0.692
BMR (kJ)	6056.8 ± 336.52	6020.04 ± 325.18	6053.5 ± 278.96	6008.69 ± 274.83	6072 ± 203.65	6057.5 ± 195.87	0.989	0.265	0.926	0.869
%FM (%)	23.83 ± 5.59	22.45 ± 5.74	24.33 ± 4.8	23.23 ± 5.24	22.65 ± 3.46	23.25 ± 2.05	0.905	0.269	0.431	0.415
FM (kg)	14.81 ± 5.24	13.83 ± 5.29	15.04 ± 4.02	14.28 ± 4.04	13.75 ± 3.18	14 ± 2.26	0.961	0.228	0.515	0.420
FFM (kg)	45.69 ± 3.28	46.15 ± 3.36	45.75 ± 2.93	45.98 ± 2.63	46.7 ± 1.56	46.1 ± 2.26	0.975	0.925	0.403	0.299
TBW (kg)	33.46 ± 2.67	33.82 ± 2.5	33.51 ± 2.14	33.69 ± 1.9	34.2 ± 1.13	33.75 ± 1.63	0.979	0.928	0.647	0.471
TC (mg/dL)	168.59 ± 24.91	167.84 ± 27.48	177.31 ± 22.47	169.06 ± 28.27	192 ± 19.8	173 ± 11.31	0.528	0.078	0.201	0.098
TGL (mg/dL)	80.12 ± 32.77	84.04 ± 36.46	81.25 ± 32.42	78.31 ± 24.42	83.5 ± 9.19	90.5 ± 16.26	0.930	0.752	0.729	0.490
HDL (mg/dL)	64.79 ± 13.2	60.83 ± 13.61	65.11 ± 13.01	60.51 ± 12.93	85.95 ± 2.9	78.25 ± 5.3	0.087	**0.048**	0.870	0.716
LDL (mg/dL)	87.7 ± 21.74	90.19 ± 23.53	95.81 ± 21.95	92.95 ± 25.87	89.05 ± 21.28	76.85 ± 19.59	0.546	0.404	0.362	0.199
Glucose (mg/dL)	78.4 ± 10.05	75.67 ± 10.39	77.88 ± 7.44	76 ± 6.23	70.5 ± 16.26	63.5 ± 16.26	0.264	0.134	0.795	0.909

Mean ± standard deviation; *p* values (ANOVA) for main effects (genotype and training) and genotype × training interaction; bold *p* values—statistically significant differences (*p* < 0.05).

**Table 4 ijerph-19-04942-t004:** Training responses by *DRD2* rs1799732 genotypes.

Parameter	DRD2 rs1799732 Genotypes				*p* Values		
	C (-) (*n* = 38)		CC (*n* = 127)		Genotype	Training	Genotype × Training
	Before Training	After Training	Before Training	After Training			
Body mass (kg)	60.78 ± 7.34	59.84 ± 7.11	60.57 ± 7.84	59.89 ± 7.76	0.956	**0.000**	0.391
BMI (kg/m^2^)	21.58 ± 2.11	21.32 ± 1.92	21.62 ± 2.55	21.4 ± 2.53	0.893	**0.000**	0.595
BMR (kJ)	6068.74 ± 310.58	6023.82 ± 311.16	6053.06 ± 335.31	6018.07 ± 321.59	0.856	**0.000**	0.636
%FM (%)	24.02 ± 4.84	22.78 ± 4.5	23.82 ± 5.68	22.47 ± 5.97	0.800	**0.000**	0.796
FM (kg)	14.9 ± 4.64	13.96 ± 4.47	14.79 ± 5.25	13.85 ± 5.34	0.911	**0.000**	0.996
FFM (kg)	45.67 ± 3.14	45.93 ± 3.18	45.71 ± 3.25	46.2 ± 3.31	0.792	**0.002**	0.332
TBW (kg)	33.44 ± 2.3	33.71 ± 2.45	33.48 ± 2.7	33.84 ± 2.43	0.844	**0.018**	0.731
TC (mg/dL)	170.45 ± 24.64	165.66 ± 25.41	169.5 ± 24.91	168.72 ± 27.92	0.811	0.154	0.302
TGL (mg/dL)	88.37 ± 43.07	83 ± 27.01	77.85 ± 28.29	83.73 ± 37.46	0.372	0.933	0.067
HDL (mg/dL)	61.69 ± 9.75	58.86 ± 11.88	66.09 ± 14.03	61.65 ± 14.01	0.114	**0.000**	0.420
LDL (mg/dL)	90.99 ± 21.26	90.19 ± 20.81	87.76 ± 21.93	90.33 ± 24.52	0.684	0.632	0.360
Glucose (mg/dL)	78.71 ± 10.27	73.87 ± 9.65	78.12 ± 9.78	76.06 ± 10.28	0.619	**0.000**	0.140

Mean ± standard deviation; *p* values (ANOVA) for main effects (genotype and training) and genotype × training interaction; bold *p* values—statistically significant differences (*p* < 0.05).

**Table 5 ijerph-19-04942-t005:** Training responses by *DRD2* rs1800497 genotypes.

Parameter	DRD2 rs1800497 Genotypes						*p* Values			
	C/T (*n* = 57)		CC (*n* = 103)		TT (*n* = 5)		Genotype	Training	Genotype × Training	Genotype × Training TT + CT vs. CC
	Before Training	After Training	Before Training	After Training	Before Training	After Training				
Body mass (kg)	60.55 ± 7.93	59.78 ± 7.75	60.64 ± 7.69	59.96 ± 7.63	60.84 ± 6.71	59.32 ± 6.13	0.993	**0.000**	0.514	0.558
BMI (kg/m^2^)	21.36 ± 2.19	21.12 ± 2.04	21.73 ± 2.57	21.52 ± 2.57	21.96 ± 2.91	21.42 ± 2.66	0.612	**0.000**	0.366	0.524
BMR (kJ)	6067.28 ± 350.35	6031.6 ± 339.09	6050.59 ± 321.28	6017.22 ± 311.69	6060.8 ± 287.41	5925 ± 230.61	0.910	**0.000**	0.140	0.569
%FM (%)	23.89 ± 5.49	22.45 ± 5.63	23.85 ± 5.4	22.63 ± 5.61	23.76 ± 8.31	21.58 ± 7.85	0.973	**0.000**	0.572	0.427
FM (kg)	14.85 ± 5.13	13.75 ± 5.15	14.81 ± 5.08	13.98 ± 5.16	14.58 ± 6.4	13.18 ± 5.84	0.973	**0.000**	0.491	0.259
FFM (kg)	45.71 ± 3.32	46.07 ± 3.24	45.7 ± 3.24	46.16 ± 3.37	45.82 ± 1.41	46.4 ± 1.52	0.989	**0.024**	0.840	0.651
TBW (kg)	33.66 ± 2.75	33.78 ± 2.36	33.37 ± 2.58	33.82 ± 2.52	33.32 ± 1.4	33.94 ± 1.3	0.951	0.079	0.352	0.217
TC (mg/dL)	168.21 ± 26.93	167.11 ± 30.24	170.92 ± 23.7	169.18 ± 25.74	162 ± 23.31	154.4 ± 25.18	0.500	0.301	0.804	0.974
TGL (mg/dL)	73.12 ± 24.7	81.65 ± 34.34	83.92 ± 35.96	84.91 ± 36.31	86.6 ± 23.14	77.6 ± 26.21	0.355	0.974	0.275	0.253
HDL (mg/dL)	66.08 ± 13.77	60.92 ± 15.05	64.18 ± 12	60.92 ± 12.51	72.12 ± 28.16	63.88 ± 19.14	0.593	**0.001**	0.387	0.217
LDL (mg/dL)	87.45 ± 21.82	89.88 ± 26.17	89.87 ± 21.84	91.27 ± 22.53	72.4 ± 13.61	75 ± 9.02	0.186	0.501	0.949	0.745
Glucose (mg/dL)	79 ± 8.49	75.05 ± 9.31	77.94 ± 10.71	76.23 ± 10.56	76.2 ± 6.69	67.2 ± 8.17	0.393	**0.003**	0.153	0.106

Mean ± standard deviation; *p* values (ANOVA) for main effects (genotype and training) and genotype × training interaction; bold *p* values—statistically significant differences (*p* < 0.05).

**Table 6 ijerph-19-04942-t006:** Training responses by *DRD2* rs1800498 genotypes.

Parameter	DRD2 rs1800498 Genotypes						*p* Values		
	C/T (*n* = 78)		CC (*n* = 34)		TT (*n* = 53)		Genotype	Training	Genotype × Training
	Before Training	After Training	Before Training	After Training	Before Training	After Training			
Body mass (kg)	61.25 ± 7.78	60.47 ± 7.78	59.63 ± 6.28	58.96 ± 6.24	60.33 ± 8.44	59.59 ± 8.15	0.575	**0.000**	0.945
BMI (kg/m^2^)	21.92 ± 2.5	21.7 ± 2.46	21.08 ± 1.78	20.87 ± 1.7	21.48 ± 2.69	21.24 ± 2.64	0.211	**0.000**	0.948
BMR (kJ)	6074.36 ± 334.49	6027.65 ± 324.7	6042.59 ± 278.89	6007.68 ± 256.45	6039.66 ± 353.95	6014.75 ± 348.48	0.885	**0.000**	0.554
%FM (%)	24.53 ± 5.58	23.09 ± 5.75	23.75 ± 5.06	22.23 ± 5.67	22.96 ± 5.56	21.92 ± 5.52	0.363	**0.000**	0.515
FM (kg)	15.41 ± 5.2	14.38 ± 5.2	14.38 ± 4.35	13.5 ± 4.71	14.24 ± 5.39	13.37 ± 5.35	0.420	**0.000**	0.844
FFM (kg)	45.79 ± 3.32	46.22 ± 3.54	45.2 ± 2.75	45.95 ± 2.53	45.91 ± 3.37	46.12 ± 3.34	0.783	**0.000**	0.160
TBW (kg)	33.66 ± 2.66	33.86 ± 2.59	33.06 ± 2.04	33.66 ± 1.85	33.45 ± 2.85	33.83 ± 2.55	0.724	**0.001**	0.384
TC (mg/dL)	171.08 ± 23.76	167.33 ± 26.88	162 ± 28.65	162.38 ± 29.85	172.66 ± 22.99	172.64 ± 25.98	0.135	0.517	0.497
TGL (mg/dL)	79.73 ± 35.01	79.18 ± 29.6	76.71 ± 29.9	84.29 ± 35.37	83.36 ± 30.33	89.55 ± 41.98	0.395	0.109	0.368
HDL (mg/dL)	65.77 ± 11.89	60.42 ± 13.39	62 ± 15.74	59.95 ± 12.89	66.02 ± 13.45	62.56 ± 14.34	0.473	**0.000**	0.290
LDL (mg/dL)	89.29 ± 21.45	91.08 ± 25.16	84.6 ± 23.78	85.64 ± 21.95	89.86 ± 20.98	92.13 ± 22.42	0.383	0.303	0.961
Glucose (mg/dL)	79.64 ± 10.17	76.65 ± 10.37	78.82 ± 8.42	73.47 ± 8.98	75.85 ± 10	75.26 ± 10.49	0.205	**0.000**	0.096

Mean ± standard deviation; *p* values (ANOVA) for main effects (genotype and training) and genotype × training interaction; bold *p* values—statistically significant differences (*p* < 0.05).

**Table 7 ijerph-19-04942-t007:** Haplotype analysis with a relative change as the dependent variable, and haplotype and baseline value as independent variables.

Parameter	C/A/C/C/C		C/A/C/T/T		C/G/C/C/C		C/G/C/C/T		A/A/C/T/C		C/A/-/C/C		C/A/-/C/T	
	14.62%		1.79%		3.27%		1.77%		15.34%		6.90%		2.54%	
	coef	*p*	coef	*p*	coef	*p*	coef	*p*	coef	*p*	coef	*p*	coef	*p*
Body mass (kg)	0.0170	0.9673	−0.2562	0.7337	1.3757	0.2163	0.6695	0.1153	1.4922	0.2560	0.5591	0.5075	0.9027	0.3630
BMI (kg/m^2^)	0.0842	0.8245	0.4673	0.4506	1.1705	0.2827	0.6478	0.0977	1.9021	0.0810	−0.0822	0.9116	0.5959	0.5129
BMR (kJ)	−0.1524	**0.0000**	0.0049	**0.0000**	0.3525	**0.0000**	0.2063	**0.0000**	−0.1967	**0.0000**	−0.2665	**0.0000**	0.2850	**0.0000**
FM (%)	−1.0129	0.5500	2.1964	0.3864	6.3480	0.1259	−0.6316	0.7126	3.6267	0.4817	0.1311	0.9695	6.0718	0.1242
Extent of fat mass (kg)	−0.8987	0.6534	2.1774	0.4781	6.2615	0.1952	1.1277	0.5759	−2.0462	0.7353	1.4107	0.7237	6.8413	0.1413
FFM (kg)	−0.0707	0.8753	−0.3929	0.5492	−1.1845	0.3160	1.1992	**0.0093**	0.1639	0.8889	−0.2564	0.7754	−1.3644	0.1861
TBW (kg)	−0.4526	0.5038	−1.1448	0.2426	0.4075	0.8098	0.6765	0.3204	−0.7086	0.6963	−0.6302	0.6468	−1.4940	0.3297
TC (mg/dL)	−1.2554	0.5265	−4.8291	0.1024	−0.2949	0.9510	0.4733	0.8127	−0.0521	0.9914	−0.5233	0.9028	−8.0054	0.0708
TGL (mg/dL)	−0.4640	0.9381	−7.0028	0.4185	−5.9037	0.6649	0.1760	0.9761	−15.4462	0.3171	−14.0280	0.2389	5.0381	0.7640
HDL (mg/dL)	−3.8704	0.1490	−0.3162	0.9365	−5.5992	0.3969	2.4488	0.3677	13.3063	**0.0497**	2.0698	0.7265	−0.2356	0.9718
LDL (mg/dL)	0.4191	0.9104	−6.6574	0.2144	2.6792	0.7631	−2.4015	0.5245	−8.9420	0.3162	1.9869	0.8058	−17.2637	**0.0455**
Glucose (mg/dL)	−4.1140	**0.0318**	−6.8617	**0.0204**	2.3896	0.6034	0.3082	0.8716	−5.1268	0.3511	−0.5601	0.8862	−8.1499	0.0773

Bold *p* values—statistically significant differences (*p* < 0.05).

## Data Availability

The data presented in this study are available on request from the corresponding author. The data are not publicly available due to privacy/ethical restrictions.

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
