# Peer review of "Impact of the DRD2 Polymorphisms on the Effectiveness of the Training Program"

_ijerph, 2022, doi:10.3390/ijerph19094942_

Round 1

Reviewer 1 Report

The study investigated the association between DRD2 polymorphic sites and the response of the body to a specific 12-week training program. The study was conducted on 165 females. Genotypes and haplotypes distribution were studied. Changes in body mass and body composition measurements, lipid profile, and glucose levels were assessed before and after a 12-week training program. 

Abstract: In addition to the p-values, please include the amount of decrease in glucose and LDL-C and increase in BMR and FFM.

Line 21 & 24: The first time you include an abbreviation in an abstract, you should write out the entire word and include the abbreviation.

Lines 38-40: Please include a reference.

Lines 92-93: This was an intervention study; therefore, the personal information and results could not have been anonymous. Please clarify.

Line 117: Could you include the adherence rates to the exercise program?

Tables 1-5: Please add a note under the table explaining the table. Are the values means ± standard deviations? Do the bold signify statistically significant at p<.05? Please also include and explanation of the p-values.

Table 6: Add information about the bolded p-values under the table.

Line 330-331: Your paper did not convince me that the DRD2 played an important role in post-training changes of lipid and carbohydrate metabolism. It might play a role in it. Also, the effectiveness of the training program depends on the goals of the program.

Line 345-346: Are you suggesting that knowing the DRD2 gene might identify people that react well or poorly to exercise? I’m unsure why you suggest individuals are reacting poorly to exercise. Are you sure it wasn’t the type of exercise that you had the individual participating in? Might some individuals respond better to different types of exercise rather than all exercise?

Reviewer 2 Report

In this manuscript, the authors examine the repercussions of several DRD2 polymorphisms on the successful outcome of physical training in a wide sample of Polish female population. The study is well conducted with a proper methodological plan, and the results are analyzed with the correct statistical methods. Moreover, the haplotype reconstruction is an element of novelty that sheds new light on the understanding of the role of polymorphisms, especially when they don't show a strong effect when taken individually. I have a small number of minor comments on the text:

  • In the abstract, the second sentence (lines 15-17) is not well related to the rest of the text, consider revising it.
  • line 61: i do not agree with the length of the DRD2 mRNA, since the locus can generate different transcript isoforms that encode for the same protein. The authors should specify it, or remove the informations on transcript length since they are not very relevant.
  • line 81: the authors introduce the methodological plan including only Caucasian females, but they explain their choice only in the discussion. I suggest adding a sentence in the introduction explaining the different outcome of DRD2 polymorphisms in females compared to males.
  • line 260: "performer" is a typo, correct in "performed"
  • lines 273-277: the sentence is not clear and not well written in English, consider revising.
  • line 291: substitute "ethical" with "ethnical".

Given the scientific evidences presented and the quality of the analysis, I would recommend this manuscript for publication in this journal.
